# Contrastive Graph Poisson Networks: Semi-Supervised Learning with Extremely Limited Labels

**Sheng Wan**[1,2,3,*]**, Yibing Zhan**[3]**, Liu Liu**[4]**, Baosheng Yu**[4]**, Shirui Pan**[5]**, Chen Gong**[1,2,3,†]

[1]PCA Lab, Key Lab of Intelligent Perception and Systems for
High-Dimensional Information of Ministry of Education
[2]Jiangsu Key Lab of Image and Video Understanding for Social Security,
School of Computer Science and Engineering, Nanjing University of Science and Technology
[3]JD Explore Academy [4]The University of Sydney
[5]Department of Data Science and AI, Faculty of IT, Monash University
`wansheng315@hotmail.com`, `zhanyibing@jd.com`,
{`liu.liu1, baosheng.yu`}`@sydney.edu.au`,
`shirui.pan@monash.edu`, `chen.gong@njust.edu.cn`

## Abstract

Graph Neural Networks (GNNs) have achieved remarkable performance in the task of semi-supervised node classification. However, most existing GNN models require sufficient labeled data for effective network training. Their performance can be seriously degraded when labels are extremely limited. To address this issue, we propose a new framework termed Contrastive Graph Poisson Networks (CGPN) for node classification under extremely limited labeled data. Specifically, our CGPN derives from variational inference; integrates a newly designed Graph Poisson Network (GPN) to effectively propagate the limited labels to the entire graph and a normal GNN, such as Graph Attention Network, that flexibly guides the propagation of GPN; applies a contrastive objective to further exploit the supervision information from the learning process of GPN and GNN models. Essentially, our CGPN can enhance the learning performance of GNNs under extremely limited labels by contrastively propagating the limited labels to the entire graph. We conducted extensive experiments on different types of datasets to demonstrate the superiority of CGPN.

## 1 Introduction

Graph-based Semi-Supervised Learning (SSL) refers to classifying unlabeled data based on a handful of labeled data and a given graph structure indicating the connections between all data. Recently, graph-based SSL has attracted increasing attention due to its solid mathematical foundation, and satisfactory performance [1, 2, 3].

As the mainstream to solve graph-based SSL problems, Graph Neural Networks (GNNs), which operate in the graph domain, have achieved impressive performance in recent years [4, 5, 6, 7]. Nevertheless, current GNNs, such as Graph Convolutional Networks (GCNs) [8] and graph attention networks (GATs) [9], require sufficient labeled data to obtain satisfactory generalization abilities. Unfortunately, the reliance on sufficient labeled data increases the burden of data collection, and the number of labels can be extremely limited in some real-world scenarios. The performance of most

---

[*]This work was done when Sheng was a research intern at JD Explore Academy.
[†]C. Gong is the corresponding author.

current GNNs seriously declines as the label size shrinks, since the scarce supervision signals are insufficient to train a model with satisfactory discriminative ability, see Figure 1. To the best of our knowledge, few studies have focused on semi-supervised classification with GNNs at extremely low label rates [10].

In line with the aforementioned observations, this paper proposes a new framework termed Contrastive Graph Poisson Networks (CGPN) to address the problem of semi-supervised node classification under extremely low label rates. Deriving from the variational inference [12, 13], our proposed CGPN framework approximates the intractable posterior with a surrogate distribution, where two types of GNNs have been adopted for instantiation.

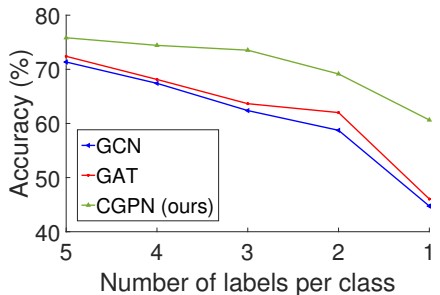

Figure 1: Classification performance of GCN, GAT, and our proposed CGPN with different sizes of labeled data on Cora [11] dataset.

Accordingly, we first design a new Graph Poisson Network (GPN) to propagate the limited labels to the entire graph effectively. Specifically, our GPN is motivated by Poisson learning [14]; flexibly models the Poisson label propagation based on attention mechanism; and leverages the structural information to better guide the propagation of labels. Meanwhile, we exploit another GNN, such as GAT, together with the proposed GPN to model the approximated posterior according to variational inference.

On this basis, we acquire predictions from two comparable views, where a contrastive objective can be naturally incorporated to jointly refine the learning process of the GPN and GNN models. Moreover, the supervision signals implicitly contained in the massive unlabeled data can be exploited with the formulated contrastive loss. As a result, the model learning ability of our proposed framework can be lifted. Experimental results on benchmark datasets confirm the strong benefits of our proposed CGPN when dealing with semi-supervised node classification at very low label rates.

In summary, the contributions of this paper lie in three folds:

First, we propose a novel GNN framework termed CGPN to solve the semi-supervised node classification with extremely limited labels. CGPN significantly outperforms the existing GNNs.

Second, we design a new Graph Poisson Network (GPN). Different from the Poisson learning algorithm, our GPN incorporates graph-structure information and could be trained in an end-to-end manner to guide the propagation of labels more flexibly.

Third, we integrate contrastive learning into the variational inference framework, so that extra supervision information can be explored from the massive unlabeled data to help train our CGPN framework.

## 2   Problem description

We start by formally introducing the problem of graph-based SSL. Given a set of $n = l + u$ examples $\Psi = \{\mathbf{x}_1, \cdots, \mathbf{x}_l, \mathbf{x}_{l+1}, \cdots, \mathbf{x}_n\}$, where the first $l$ examples are provided with the labels $\{y_i\}_{i=1}^l$ and the remaining $u$ examples constitute the unlabeled set with typically $l \ll u$. Let $\mathbf{X} \in \mathbb{R}^{n \times d}$ denote the feature matrix with the $i$-th row formed by the feature vector $\mathbf{x}_i$, and $\mathbf{Y} \in \mathbb{R}^{n \times c}$ denote the label matrix with its $(i, j)$-th element $\mathbf{Y}_{ij} = 1$ if $\mathbf{x}_i$ belongs to the $j$-th class and $0$ otherwise. Here $d$ is the feature dimension and $c$ is the number of classes. The dataset $\Phi$ is represented by an undirected graph $\mathcal{G} = \langle \mathcal{V}, \mathcal{E} \rangle$, where $\mathcal{V}$ represents the node set containing all examples and $\mathcal{E}$ is the edge set modeling the similarity among the nodes/examples. The adjacency matrix of $\mathcal{G}$ is denoted as $\mathbf{A}$, where $\mathbf{A}_{ij} = 1$ if there exists an edge between $\mathbf{x}_i$ and $\mathbf{x}_j$ and $\mathbf{A}_{ij} = 0$, otherwise. In this paper, we target transductive graph-based SSL which aims to find the labels $y_{l+1}, y_{l+2}, \cdots, y_n$ of the unlabeled examples $\mathbf{x}_{l+1}, \mathbf{x}_{l+2}, \cdots, \mathbf{x}_n$ based on $\Psi$.

## 3 Related work

### 3.1 Graph-based semi-supervised learning

SSL methods focus on training models with small amounts of labeled data as well as relatively large amounts of unlabeled data for [15, 16, 17, 18, 19]. Graph-based SSL algorithms have been one of the most popular research topics during the past decades. The early graph-based techniques are designed based on the simple assumption that nearby nodes are likely to have the same label. This goal can be achieved through the low-dimensional embeddings with Laplacian eigenmaps [20, 21], Markov random walks [22], *etc.* Meanwhile, graph partition [23] offers another important line in graph-based SSL. To further enhance the learning capacities, various techniques have been proposed to model the data features and graph structure jointly, such as Planetoid [24], where the supervised classifier is regularized with a Laplacian regularizer or an embedding-based regularizer. Recently, a set of graph-based SSL approaches have been proposed to improve the performance of the above-mentioned techniques, including [25, 14, 26].

### 3.2 Graph neural networks

In the past few years, increasing attention has been paid to GNN models [27, 28, 26, 29, 30, 31, 32]. Early-staged works aim to derive diverse types of graph convolution in spectral-domain based on the graph spectral theory [8, 33, 4]. In [33], a general graph convolution framework based on graph Laplacian is first proposed. Afterwards, Defferrard *et al.* [4] approximate the convolutional filter via using a $K$-order Chebyshev polynomial, in order to avoid the intense calculations of eigendecomposition of the normalized graph Laplacian. In addition to this, Kipf and Welling [8] further simplify the graph convolution by a localized first-order approximation, which brings about more efficient filtering operations than spectral CNNs. Another line of research efforts focus on directly performing graph convolution in the spatial domain [5, 27, 34, 35]. In spatial GNN models, the convolution operation is defined as a weighted average function over the neighbors of each node, which characterizes the impact exerting to the target node from its neighboring ones. For instance, Hamilton *et al.* [5] propose a general inductive framework called GraphSAGE, which is able to learn an embedding function generalizing to previously unseen graph nodes. Besides, in [9], the graph attention network (GAT) is devised by utilizing the attention mechanism, which assigns different weights to the neighboring nodes and aggregates feature with discrimination. Although these models exploit the inter-dependencies among labeled and unlabeled nodes, their performances can still degrade dramatically when the number of labels is extremely limited.

### 3.3 Poisson learning

Poisson learning is motivated by the need to address the degeneracy of Laplacian SSL when label information is very limited. To be concrete, traditional Laplacian learning algorithm [15] aims at solving the following problem:

$$\begin{cases} \mathcal{L}u(\mathbf{x}_i) = 0, & \text{if } l+1 \le i \le n, \\ u(\mathbf{x}_i) = y_i, & \text{if } 1 \le i \le l, \end{cases} \tag{1}$$

where $\mathcal{L}$ represents the unnormalized graph Laplacian and $u(\mathbf{x}_i) \in \mathbb{R}^c$ is the label vector of $\mathbf{x}_i$. Here, the node $\mathbf{x}_i$ belongs to the $j$-th class if the $j$-th component $u_j(\mathbf{x}_i)$ is the largest in $u(\mathbf{x}_i)$. Although Laplacian learning works very well for SSL tasks with a moderate number of labeled examples, the performance becomes quite poor at very low label rates.

Different from Laplacian learning, where labels are imposed as boundary conditions, in Poisson learning, labels appear as the source term. The solution of a Poisson equation can be computed as

$$\mathcal{L}u(\mathbf{x}_i) = \sum_{j=1}^{l} (y_j - \bar{y})\delta_{ij} \text{ for } i = 1, ..., n \tag{2}$$

satisfying $\sum_{i=1}^{n} d_i u(\mathbf{x}_i) = 0$, where $\delta_{ij} = 1$ if $i = j$ and $\delta_{ij} = 0$, otherwise, $\bar{\mathbf{y}} = \frac{1}{l}\mathbf{F}\mathbf{1}$ with $\mathbf{F} \in \mathbb{R}^{c \times l}$ denoting the label matrix of the $l$ labeled nodes, and $d_i = \sum_{j=1}^{n} \mathbf{A}_{ij}$. The Poisson equation (2) can be solved efficiently with a simple iteration, and the result of the $(t+1)$-th iteration can be obtained as

$$\mathbf{U}^{(t+1)} \leftarrow \mathbf{U}^{(t)} + \mathbf{D}^{-1}(\mathbf{B}^\top - \mathbf{L}\mathbf{U}^{(t)}), \tag{3}$$

where $\mathbf{D}$ is the diagonal matrix with $\mathbf{D}_{ii} = d_i$, $\mathbf{L}$ is the Laplacian matrix, and $\mathbf{B} = [\mathbf{F} - \bar{\mathbf{y}}, \mathbf{O}]$ is the source term. Here, $\mathbf{O} \in \mathbb{R}^{c \times (n-l)}$ indicates a zero matrix. After $T$ iterations, the prediction can be acquired as $\mathbf{U}^{(T)}$. The specific interpretation of Poisson learning algorithm can be found in [14].

# 4 Methodology

This section details our proposed CGPN framework. Specifically, we describe the critical components of CGPN by explaining the variational inference framework, presenting the instantiation with GPN and GNN models, and illustrating the contrastive label inference.

## 4.1 Inference framework

To infer the labels of the unlabeled nodes, *i.e.*, $\mathbf{Y}_U$, we need to estimate the posterior distribution given the node features $\mathbf{X}$, the observed labels $\mathbf{Y}_L$, and the adjacency matrix $\mathbf{A}$, namely $p_\theta(\mathbf{Y}_U | \mathbf{A}, \mathbf{X}, \mathbf{Y}_L)$ with parameters $\theta$. Computation of this posterior is usually analytically intractable, so we resort to approximate posterior inference methods. Inspired by the recent advances in scalable variational inference [36, 37], we introduce a distribution $q_\phi(\mathbf{Y}_U | \mathbf{A}, \mathbf{X}, \mathbf{Y}_L)$ parameterized by $\phi$ to approximate the true posterior $p_\theta(\mathbf{Y}_U | \mathbf{A}, \mathbf{X}, \mathbf{Y}_L)$. Afterwards, we can write the Evidence Lower BOund (ELBO) as

$$\mathcal{L}_{ELBO}(\theta, \phi) = \log p_\theta(\mathbf{Y}_L | \mathbf{A}, \mathbf{X}) - \mathcal{D}_{\mathrm{KL}}(q_\phi(\mathbf{Y}_U | \mathbf{A}, \mathbf{X}, \mathbf{Y}_L) || p_\theta(\mathbf{Y}_U | \mathbf{A}, \mathbf{X})), \quad (4)$$

where $\mathcal{D}_{\mathrm{KL}}(\cdot || \cdot)$ represents the Kullback-Leibler divergence between two distributions. For practical use, it remains to specify the parametric forms of $q_\phi(\mathbf{Y}_U | \mathbf{A}, \mathbf{X}, \mathbf{Y}_L)$ and $p_\theta(\mathbf{Y} | \mathbf{A}, \mathbf{X})$ with GNNs.

## 4.2 Instantiations

In this section, the instantiation of $q_\phi(\mathbf{Y} | \mathbf{A}, \mathbf{X}, \mathbf{Y}_L)$ and $p_\theta(\mathbf{Y} | \mathbf{A}, \mathbf{X})$ will be illustrated.

### 4.2.1 Instantiation of $q_\phi(\mathbf{Y} | \mathbf{A}, \mathbf{X}, \mathbf{Y}_L)$ with Graph Poisson Networks

To approximate the posterior model $q_\phi(\mathbf{Y} | \mathbf{A}, \mathbf{X}, \mathbf{Y}_L)$, we need a strong function with the inputs $\mathbf{A}$, $\mathbf{X}$, and $\mathbf{Y}_L$ and outputs the probability of $\mathbf{Y}$. Due to the extremely scarce label information in $\mathbf{Y}_L$, most existing methods are ineffective here. Fortunately, the Poisson learning algorithm [14] is recently proposed to address the scenarios with very limited labels. The superiority of Poisson learning over traditional Laplacian learning has been proven both theoretically and experimentally at very low label rates [14]. However, the graph structure has not been fully leveraged to guide the propagation of labels in Poisson learning. Concretely, Poisson learning relies on a fixed graph which can be noisy in reality, and thus the intrinsic relationships among graph nodes cannot be well explored. Meanwhile, the structural information constituted by the neighboring node features has not been exploited, since Poisson learning mainly emphasizes the propagation of the input label information. As a consequence, inaccurate label predictions can be accumulated with iterative propagation, which inevitably results in performance degradation. To handle these difficulties, we propose a more flexible GNN model called 'Graph Poisson Networks' (GPN).

Inspired by GAT [9], we intend to adaptively capture the importance of the neighbors exerting to the target node via attention mechanism. In this way, the graph information can be gradually refined via network training, which makes the propagation of labels more reasonable. To be specific, we first compute the attention coefficient $e_{ij}$ between nodes $\mathbf{x}_i$ and $\mathbf{x}_j$ as

$$e_{ij} = \vec{\mathbf{a}}^\top [\mathbf{W}\mathbf{x}_i; \mathbf{W}\mathbf{x}_j], \quad (5)$$

where $\vec{\mathbf{a}}$ is a trainable weight vector, $\mathbf{W}$ is a trainable weight matrix, and $[\cdot; \cdot]$ denotes the concatenation operation. The attention coefficient $e_{ij}$ is usually normalized across the neighbors of $\mathbf{x}_i$ with a softmax function to make it comparable across nodes:

$$\alpha_{ij} = \frac{\exp(e_{ij})}{\sum_{k \in N_i} \exp(e_{ik})}, \quad (6)$$

where $N_i$ denotes the indices of $\mathbf{x}_i$'s neighbors. In our GPN, the normalized attention coefficient $\alpha_{ij}$ is adopted to represent the edge weights of the input adjacency matrix $\mathbf{A}$ and can be further optimized

via network training. In this way, improved edge weights can help guide the propagation of labels in a more reasonable way. At this point, the output of our GPN can be obtained by the following iteration:

$$\widetilde{\mathbf{U}}^{(t)} \leftarrow \widetilde{\mathbf{U}}^{(t-1)} + \widetilde{\mathbf{D}}^{-1} \left( \mathbf{B}^{\top} - \widetilde{\mathbf{L}}\widetilde{\mathbf{U}}^{(t-1)} \right), \tag{7}$$

where $\widetilde{\mathbf{D}}$ and $\widetilde{\mathbf{L}}$ represent the diagonal matrix and Laplacian matrix of the attention-based graph, respectively, and $\widetilde{\mathbf{U}}^{(t)}$ is the result of the $t$-th iteration.

Note that Eq. (7) fails to incorporate the structural information formed by the node features, as the prediction in each iteration mainly relies on the label information. To capture the meaningful structural information, a feature transformation module $f_{FT}$ is adopted to introduce feature information explicitly. Specifically, the feature transformation module $f_{FT}$ predicts labels based on the node features and can be expressed as a single-layer perceptron. Afterwards, by propagating the predictions of $f_{FT}$ iteratively, the feature information within the neighborhood structure can be incorporated correspondingly and further improve the final predictions. Therefore, we modify the output of the $(T-2)$-th iteration, namely $\widetilde{\mathbf{U}}^{(T-2)}$, to

$$\widetilde{\mathbf{U}}^{(T-2)} \leftarrow \widetilde{\mathbf{U}}^{(T-2)} + f_{FT}(\mathbf{X}), \tag{8}$$

where $T$ denotes the number of iterations. By doing so, the involved feature information can be further propagated and the extracted structural information can help refine the label predictions. In addition, the feature transformation module can also accelerate the convergence of the iterative process shown in Eq. (7). Note that the outcomes of $f_{FT}$ can only be propagated in the last two iterations to avoid performance degradation which is caused by over-smoothing.

### 4.2.2 Instantiation of $p_{\theta}(\mathbf{Y}|\mathbf{A}, \mathbf{X})$ with graph neural networks

For $p_{\theta}(\mathbf{Y}|\mathbf{A}, \mathbf{X})$ in the ELBO obejective function, we can flexibly instantiate it with a GNN model that takes $\mathbf{A}$ and $\mathbf{X}$ as the inputs and outputs the probability of $\mathbf{Y}$. Note that most of the existing GNN models are applicable. In this paper, the well-known Graph Convolutional Networks (GCN) [8] and GAT [9] are considered for instantiation. Note that when we employ GAT to instantiate $p_{\theta}(\mathbf{Y}|\mathbf{A}, \mathbf{X})$, the attention coefficients are shared across GPN and GAT, so that the scale of network parameters can be reduced. Additionally, the GAT can help guide the propagation of labels through the shared attention coefficients.

## 4.3 Contrastive label inference

Furthermore, we intend to leverage the supervision signals beyond the limited labels. In this paper, contrastive learning [38, 39] is utilized to explore extra supervision information from the massive unlabeled data for model training, which can improve the performance of label inference. To be specific, we maximize the agreement between the predictions of the same node that are generated from $q_{\phi}(\mathbf{Y}|\mathbf{A}, \mathbf{X}, \mathbf{Y}_L)$ and $p_{\theta}(\mathbf{Y}|\mathbf{A}, \mathbf{X})$, *i.e.*, $\mathbf{z}_i$ and $\tilde{\mathbf{z}}_i$. Meanwhile, we pull the predictions of different node pairs away. As a result, the pairwise contrastive loss between $\mathbf{z}_i$ and $\tilde{\mathbf{z}}_i$ can be defined as

$$\mathcal{L}_{PC}(\mathbf{z}_i, \widetilde{\mathbf{z}}_i) = -\log \frac{\exp(\langle \mathbf{z}_i, \widetilde{\mathbf{z}}_i \rangle / \tau)}{\exp(\langle \mathbf{z}_i, \widetilde{\mathbf{z}}_i \rangle / \tau) + \sum_{j=1}^{n} \mathbb{1}_{[j \neq i]} \exp(\langle \mathbf{z}_i, \widetilde{\mathbf{z}}_j \rangle / \tau) + \sum_{j=1}^{n} \mathbb{1}_{[j \neq i]} \exp(\langle \mathbf{z}_i, \mathbf{z}_j \rangle / \tau)}, \tag{9}$$

where $\langle \cdot, \cdot \rangle$ denotes the inner product and $\tau$ is a tunable temperature parameter. Based on Eq. (9), the overall contrastive objective to be minimized is

$$\mathcal{L}_{Cont} = \frac{1}{2n} \sum_{i=1}^{n} \left( \mathcal{L}_{PC}(\mathbf{z}_i, \tilde{\mathbf{z}}_i) + \mathcal{L}_{PC}(\tilde{\mathbf{z}}_i, \mathbf{z}_i) \right). \tag{10}$$

In addition to the contrastive loss, a standard multiclass softmax cross-entropy loss $\mathcal{L}_{CE}$ should also be applied to penalize the difference between the outcomes of $q_{\phi}(\mathbf{Y}|\mathbf{A}, \mathbf{X}, \mathbf{Y}_L)$ and the ground-truth labels, *i.e.*, $\mathbf{Z}_L$ and $\mathbf{Y}_L$. Hence, by assigning the weight hyperparameters $\lambda_1$ and $\lambda_2$ to $\mathcal{L}_{CE}$ and $\mathcal{L}_{Cont}$ correspondingly, we arrive at the total loss as

$$\mathcal{L}(\theta, \phi) = -\mathcal{L}_{ELBO}(\theta, \phi) + \lambda_1 \mathcal{L}_{CE}(\mathbf{Z}_L, \mathbf{Y}_L) + \lambda_2 \mathcal{L}_{Cont}, \tag{11}$$

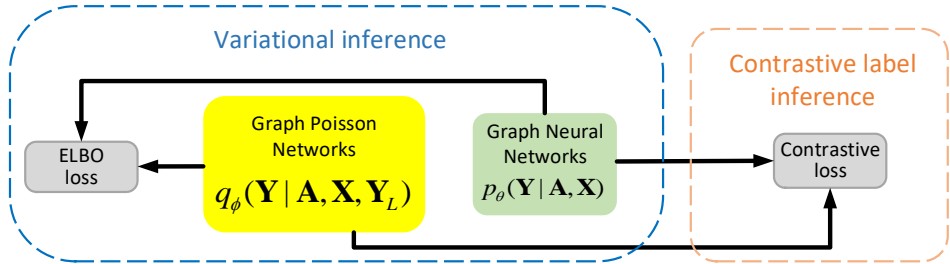

Figure 2: The framework of our Contrastive Graph Poisson Networks (CGPN). $q_\phi(\mathbf{Y}|\mathbf{A}, \mathbf{X}, \mathbf{Y}_L)$ and $p_\theta(\mathbf{Y}|\mathbf{A}, \mathbf{X})$ are instantiated by GPN and GNN models, respectively. To approximate the true posterior, the GPN and GNN models are jointly optimized based on the ELBO objective, where a contrastive loss is also utilized to explore extra supervision signals for stable model training.

Table 1: Dataset statistics

| Datasets | Nodes | Edges | Features | Classes |
|---|---|---|---|---|
| Cora | 2,708 | 5,429 | 1,433 | 7 |
| CiteSeer | 3,327 | 4,732 | 3,703 | 6 |
| PubMed | 19,717 | 44,338 | 500 | 3 |
| Amazon Photo | 7,650 | 119,081 | 745 | 8 |

where $\mathbf{Z}_L$ is generated by the instantiated model $q_\phi(\mathbf{Y}|\mathbf{A}, \mathbf{X}, \mathbf{Y}_L)$ denoting the prediction of the labeled examples $\mathbf{X}_L$. With the contrastive objective across the GPN and GNN models, the mutual information can be exploited to improve the ability of label inference. The overall framework is referred to as Contrastive Graph Poisson Networks (CGPN), of which the overview is exhibited in Figure 2.

## 5 Experiments

To reveal the effectiveness of our proposed CGPN framework, extensive experiments have been conducted on the task of semi-supervised node classification. We mainly focus on evaluating the model performance under label-scarce settings.

### 5.1 Experimental settings

**Datasets.** The experiments are conducted on four commonly used benchmark datasets, including three widely-used citation networks (*i.e.*, Cora, CiteSeer, and PubMed) [11, 40], and one Amazon product co-purchase networks (*i.e.*, Amazon Photo) [41]. In the citation networks, the nodes represent documents and their links refer to citations between documents, where each node is associated with a bag-of-words feature vector and a ground-truth label. In the product co-purchase networks, the nodes represent goods and the links indicate that two goods are frequently bought together, where node features are bag-of-words encoded product reviews, and class labels correspond to the product categories. The dataset statistics are summarized in Table 1.

**Baselines.** When we evaluate the performance of different methods in the label-scarce settings, six state-of-the-art models are used for comparison, including GCN [8], GAT [9], Bayesian Graph Convolutional Neural Networks (BGCN) [42], Multi-View Graph Representation Learning (MVGRL) [43], Generalized PageRank Graph Neural Networks (GPRGNN) [44], and Approximate Personalized Propagation of Neural Predictions (APPNP) [45]. BGCN enhances the learning ability of GNNs by modeling the uncertainty of the graph structure. MVGRL explores the supervision information contained in the unlabeled data by contrasting the encodings from two structural views of graphs. GPRGNN jointly optimizes node feature and topological information extraction to obtain excellent learning performance for label patterns. APPNP leverages a large and adjustable neighborhood for convolution by using the relationship between GCN and PageRank. Besides, we also include the Multi-Layer Perceptron (MLP) without using any graph information as a competitor, where the

Table 2: Classification accuracy with different label rates on Cora dataset

| # Labels per class | 1 | 2 | 3 | 4 |
|---|---|---|---|---|
| MLP | 26.25±3.05 | 32.19±3.94 | 36.20±3.73 | 38.92±2.59 |
| GCN [8] | 44.73±9.16 | 58.73±7.29 | 62.38±5.59 | 67.41±4.09 |
| GAT [9] | 46.04±8.07 | 62.01±7.29 | 63.65±6.04 | 68.13±5.07 |
| BGCN [42] | 49.92±8.72 | 64.55±6.84 | 64.98±6.61 | 71.69±6.62 |
| MVGRL [43] | 56.02±7.04 | 68.30±4.86 | 71.39±5.08 | 73.79±4.29 |
| GPRGNN [44] | 51.65±11.61 | 62.56±6.36 | 68.54±7.28 | 71.69±6.62 |
| APPNP [45] | 53.52±12.05 | 62.07±4.46 | 62.02±9.34 | 70.92±3.79 |
| CGPN-GCN | 60.64±9.18 | 69.15±9.03 | **73.54±2.76** | 74.43±2.25 |
| CGPN-GAT | **61.17±7.77** | **69.93±7.01** | 73.19±4.35 | **75.60±1.65** |

Table 3: Classification accuracy with different label rates on CiteSeer dataset

| # Labels per class | 1 | 2 | 3 | 4 |
|---|---|---|---|---|
| MLP | 26.10±5.00 | 30.34±5.74 | 35.99±5.42 | 38.44±4.60 |
| GCN [8] | 32.00±9.77 | 43.11±4.89 | 50.44±5.88 | 56.14±3.24 |
| GAT [9] | 35.10±8.77 | 44.74±9.01 | 53.68±5.75 | 59.43±1.88 |
| BGCN [42] | 35.16±8.04 | 46.48±5.70 | 55.49±6.97 | 58.99±5.26 |
| MVGRL [43] | 42.65±7.89 | 56.66±5.78 | 61.70±3.70 | 63.70±2.33 |
| GPRGNN [44] | 32.30±11.41 | 46.38±9.46 | 52.60±5.14 | 59.59±4.30 |
| APPNP [45] | 47.94±10.46 | 56.59±9.22 | 58.61±11.04 | 62.34±5.47 |
| CGPN-GCN | 50.49±9.72 | 58.45±7.05 | **62.07±3.76** | 64.79±2.11 |
| CGPN-GAT | **52.68±9.25** | **58.52±6.16** | 62.02±3.88 | **65.21±2.76** |

number of layers is set to two. For the proposed CGPN framework, we implement two model variants where GCN and GAT are used for instantiation, namely CPGN-GCN and CGPN-GAT.

**Training details.** For all the adopted datasets, we randomly choose one, two, three, and four labeled nodes per class for training, respectively, in order to evaluate the model performance under label-scarce settings. The hyperparameters, such as the number of hidden units and the learning rate, are determined via grid search. In our experiments, the original architecture of GCN is adopted in both the baselines and CGPN-GCN. In CGPN-GAT, the attention coefficients are shared between GPN and GAT, where only the single-head attention mechanism is utilized for simplicity. The experiments are conducted on a Linux server equipped with a Tesla P40 GPU.

## 5.2 Node classification results

Here, we present the classification results of our proposed CGPN framework (CGPN-GCN and CGPN-GAT) and the baseline methods at different label rates. The experimental results on Cora, CiteSeer, PubMed, and Amazon Photo datasets are shown in Tables 2, 3, 4, and 5, respectively, where the highest record at each label rate is highlighted in bold. We observe that both CGPN-GCN and CGPN-GAT achieve substantial performance gains at different label rates when compared with the baselines. In particular, the margin between our proposed framework and the best baseline method can exceed 4% on Cora and CiteSeer datasets given one labeled node per class, which demonstrates that the CGPN framework could effectively enhance the learning performance of GNNs. Although MVGRL utilizes the contrastive objective for graph representation learning, our proposed CGPN-GCN and CGPN-GAT exhibit better classification results. We hypothesize that the gap between MVGRL and our CGPN in accuracy is mainly due to that the devised GPN transfers as much knowledge as possible from the limited labeled nodes to the massive unlabeled ones. Notably, we could also find that the performance of the two variants, *i.e.*, CGPN-GCN and CGPN-GAT, seems quite similar since CGPN is a flexible framework and is insensitive to the choices of GNN instantiations.

Table 4: Classification accuracy with different label rates on PubMed dataset

| # Labels per class | 1 | 2 | 3 | 4 |
|---|---|---|---|---|
| MLP | 47.84±5.53 | 53.24±4.73 | 57.09±4.82 | 60.29±1.81 |
| GCN [8] | 55.40±6.69 | 62.31±5.89 | 64.48±5.57 | 69.34±4.19 |
| GAT [9] | 56.49±7.81 | 63.03±5.77 | 64.43±5.19 | 68.47±2.98 |
| BGCN [42] | 57.97±6.56 | 63.86±7.41 | 66.09±3.86 | 68.10±3.20 |
| MVGRL [43] | 52.20±12.93 | 60.79±8.74 | 64.51±8.06 | 67.69±7.67 |
| GPRGNN [44] | 58.36±10.20 | 61.30±6.59 | 64.77±8.05 | 69.73±4.50 |
| APPNP [45] | 48.89±15.50 | 63.74±7.55 | 66.82±9.76 | 69.27±4.70 |
| CGPN-GCN | 59.84±6.76 | 64.60±3.20 | 67.01±4.05 | **70.94±4.11** |
| CGPN-GAT | **61.85±5.60** | **65.18±3.87** | **68.67±4.36** | 70.58±3.87 |

Table 5: Classification accuracy with different label rates on Amazon Photo dataset

| # Labels per class | 1 | 2 | 3 | 4 |
|---|---|---|---|---|
| MLP | 36.84±9.12 | 42.67±5.17 | 55.96±7.02 | 55.54±5.39 |
| GCN [8] | 67.80±10.17 | 78.42±7.42 | 82.53±5.07 | 83.37±2.61 |
| GAT [9] | 60.68±12.93 | 74.59±6.96 | 78.70±3.78 | 81.64±3.04 |
| BGCN [42] | 53.69±14.43 | 69.76±12.67 | 78.26±8.23 | 79.66±4.49 |
| MVGRL [43] | 59.71±8.29 | 71.35±7.12 | 75.56±5.29 | 76.74±4.63 |
| GPRGNN [44] | 64.92±12.75 | 75.28±8.20 | 81.59±3.10 | 82.10±2.71 |
| APPNP [45] | 67.11±8.40 | 70.03±11.51 | 78.99±3.23 | 81.17±4.03 |
| CGPN-GCN | 71.17±8.94 | **79.54±5.18** | 84.09±3.98 | 84.43±1.84 |
| CGPN-GAT | **71.35±11.25** | 78.58±5.48 | **84.57±2.24** | **85.41±1.89** |

## 5.3 Parametric sensitivity

There are two important hyperparameters that should be manually tuned in our objective function Eq. (11). The first is $\lambda_1$ used to regularize the cross-entropy loss of GPN. Another is $\lambda_2$ assigned to the contrastive objective. By varying $\lambda_1$ and $\lambda_2$ from 0.1 to 1.5 with an interval of 0.2, the corresponding parametric sensitivity is shown in Figure 3. We find that the behavior of the proposed CGPN framework is relatively stable with the change of $\lambda_1$ and $\lambda_2$. We speculate that GPN and contrastive modules are mutually beneficial and can collaborate to obtain promising results. Generally, the two hyperparameters are not difficult to tune in practical applications.

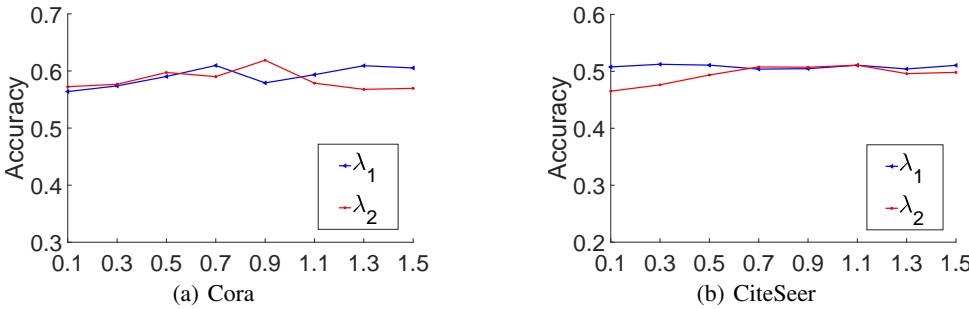

Figure 3: Parametric sensitivity of $\lambda_1$ and $\lambda_2$.

## 5.4 Ablation study

Inspired by the Poisson Learning (PL) algorithm, the devised GPN model learns to propagate the limited labels to the entire graph with structural information in an end-to-end way. To shed light on this critical component, we report the classification results obtained by using GPN and PL in Table 6 and 7, where the sizes of labeled nodes per class are kept identical to those in Sec. 5.2. The results

Table 6: Classification accuracy of reduced models on Cora dataset

| # Labels per class | 1 | 2 | 3 | 4 |
|---|---|---|---|---|
| PL [14] | 46.23±7.96 | 60.84±6.50 | 63.24±5.84 | 65.59±5.36 |
| GPN | 49.26±8.66 | 64.45±6.09 | 66.90±5.85 | 67.40±5.81 |
| CGPN-GCN | 60.64±9.18 | 69.15±9.03 | **73.54±2.76** | 74.43±2.25 |
| CGPN-GAT | **61.17±7.77** | **69.93±7.01** | 73.19±4.35 | **75.60±1.65** |

Table 7: Classification accuracy of reduced models on Citeseer dataset

| # Labels per class | 1 | 2 | 3 | 4 |
|---|---|---|---|---|
| PL [14] | 36.43±7.25 | 40.69±4.87 | 43.44±5.39 | 45.33±1.85 |
| GPN | 37.64±7.70 | 48.24±7.73 | 51.27±5.56 | 53.40±4.62 |
| CGPN-GCN | 50.49±9.72 | 58.45±7.05 | **62.07±3.76** | 64.79±2.11 |
| CGPN-GAT | **52.68±9.25** | **58.52±6.16** | 62.02±3.88 | **65.21±2.76** |

show that our GPN model can effectively boost the performance of Poisson learning at extremely limited label rates. Meanwhile, the margin between GPN and CGPN frameworks cannot be ignored, which reveals the power of the extra supervision information explored from massive unlabeled data.

## 6 Conclusion

In this paper, we propose a novel CGPN framework for semi-supervised node classification with extremely limited labels. To alleviate the performance degeneracy of the existing GNNs, we devise a new model termed GPN, which can flexibly propagate the limited labels to the entire graph by exploiting the structural information. Meanwhile, a contrastive objective is employed to extract supervision information from massive unlabeled data. As a consequence, the performance of our CGPN can be effectively enhanced by optimizing the overall objective function. Experimental results reveal the superiority of our method when compared with various baseline methods. Note that how to generalize our proposed method to inductive settings will be an interesting focus of study in the future.

## 7 Broader impact

Our work could have the following positive impacts: (1) This paper provides a new framework for SSL on graph-structured data. (2) Our proposed framework can improve the performance of GNNs, especially when labeled data are extremely limited. (3) The proposed framework is compatible with different types of GNNs (e.g., GAT and GCN).

Similar to many other machine learning algorithms, the proposed framework can be used for good and also be used for harm at the same time. Note that GNNs and our method are not immune to such misuse. Although we have no optimal solution to such a problem, we believe that this can be solved in the future.

To sum up, we believe our proposed work can be beneficial to society since many important real-world applications stand to benefit from CGPN when label information is very limited.

## 8 Acknowledgments

This research was supported in part by NSF of China (Nos: 61973162 and 62002090), NSF of Jiangsu Province (No: BZ2021013), the Fundamental Research Funds for the Central Universities (Nos: 30920032202, 30921013114), CCF-Tencent Open Fund (No: RAGR20200101), Hong Kong Scholars Program (No: XJ2019036), and Australian Research Council project (Nos: DP-180103424 and FL-170100117).

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
