# Supplementary Material for "Contrastive Graph Poisson Networks: Semi-Supervised Learning with Extremely Limited Labels"

**Sheng Wan**[1,2,3,*]**, Yibing Zhan**[3]**, Liu Liu**[4]**, Baosheng Yu**[4]**, Shirui Pan**[5]**, Chen Gong**[1,2,3,†]
[1]PCA Lab, Key Lab of Intelligent Perception and Systems for
High-Dimensional Information of Ministry of Education
[2]Jiangsu Key Lab of Image and Video Understanding for Social Security,
School of Computer Science and Engineering, Nanjing University of Science and Technology
[3]JD Explore Academy [4]The University of Sydney
[5]Department of Data Science and AI, Faculty of IT, Monash University
wansheng315@hotmail.com, zhanyibing@jd.com,
{liu.liu1, baosheng.yu}@sydney.edu.au,
shirui.pan@monash.edu, chen.gong@njust.edu.cn

## 1 Implementation details

All experiments were conducted on a Linux server with a Tesla P40 GPU. Our Contrastive Graph Poisson Network (CGPN) was implemented via PyTorch 1.4.0 [1]. We adopted the Adam optimizer [2] for training. The graph attention networks (GATs) of CGPN-GAT only utilized a single-head attention mechanism for simplicity. The number of GAT layers was set as two. Other hyperparameters were adjusted based on the corresponding datasets. Tables 1, 2, 3, and 4 provide the details of the important hyperparameters.

Table 1: Parameter settings on Cora dataset

| # Labels per class | $\lambda_1$ | $\lambda_2$ | Learning rate | Training epochs | Hidden units | Dropout rate |
|---|---|---|---|---|---|---|
| 1 | 0.9 | 1.3 | 0.1 | 500 | 128 | 0.8 |
| 2 | 0.1 | 2.0 | 0.1 | 1500 | 64 | 0.8 |
| 3 | 0.6 | 1.5 | 0.1 | 1500 | 64 | 0.8 |
| 4 | 0.6 | 1.5 | 0.1 | 1500 | 64 | 0.8 |
| 5 | 0.1 | 0.9 | 0.1 | 1500 | 64 | 0.8 |

Table 2: Parameter settings on CiteSeer dataset

| # Labels per class | $\lambda_1$ | $\lambda_2$ | Learning rate | Training epochs | Hidden units | Dropout rate |
|---|---|---|---|---|---|---|
| 1 | 0.5 | 1.1 | 0.0001 | 2000 | 512 | 0.8 |
| 2 | 1.6 | 1.1 | 0.0001 | 2000 | 512 | 0.8 |
| 3 | 1.1 | 0.5 | 0.0001 | 2000 | 512 | 0.8 |
| 4 | 1.1 | 0.5 | 0.0001 | 2000 | 512 | 0.8 |
| 5 | 0.3 | 0.5 | 0.0001 | 2000 | 512 | 0.7 |

---
[*]This work was done when Sheng was a research intern at JD Explore Academy.
[†]C. Gong is the corresponding author.

Table 3: Parameter settings on PubMed dataset

| # Labels per class | $\lambda_1$ | $\lambda_2$ | Learning rate | Training epochs | Hidden units | Dropout rate |
|---|---|---|---|---|---|---|
| 1 | 1.0 | 1.0 | 0.01 | 500 | 64 | 0.4 |
| 2 | 1.0 | 1.0 | 0.01 | 500 | 64 | 0.4 |
| 3 | 0.1 | 1.0 | 0.1 | 1000 | 64 | 0.4 |
| 4 | 1.0 | 1.0 | 0.1 | 1000 | 64 | 0.4 |
| 5 | 1.0 | 1.0 | 0.1 | 1000 | 64 | 0.4 |

Table 4: Parameter settings on Amazon Photo dataset

| # Labels per class | $\lambda_1$ | $\lambda_2$ | Learning rate | Training epochs | Hidden units | Dropout rate |
|---|---|---|---|---|---|---|
| 1 | 0.1 | 0.5 | 0.1 | 500 | 64 | 0.2 |
| 2 | 0.1 | 0.5 | 0.1 | 1000 | 128 | 0.0 |
| 3 | 0.7 | 0.5 | 0.1 | 1000 | 512 | 0.6 |
| 4 | 0.7 | 0.5 | 0.1 | 1000 | 512 | 0.6 |
| 5 | 0.7 | 0.5 | 0.1 | 1000 | 512 | 0.8 |

## 2 More performance comparison

We report the classification results of our CGPN framework and the baseline methods on the four datasets when five labels per class are available for training. The statistics are listed in Table 5, where the highest record at each label rate is highlighted in bold. We can observe that CGPN still outperforms the baseline methods. The performance margin between CGPN and other competitors demonstrates the good capability of our proposed framework in semi-supervised learning with very limited labels.

In addition, the impact of the number of labeled nodes on classification performance has been exhibited in Figure 1. We can observe that the performance of all the methods would be improved with the increase of labeled data. Intuitively, CGPN could significantly improve the performance of GCN and GAT when labeled data are extremely limited. Nevertheless, the improvements achieved by CGPN are gradually shrinking as the number of labeled data increases. In particular, when there are ten labeled data per class, the performance of all methods becomes similar. Therefore, one of our future works is to enhance CGPN so that it could still significantly improve the performance of GCN or GAT when labeled data are sufficient.

## 3 Detailed descriptions of all baselines

In this paper, six baselines are used for comparison to evaluate the performance of our CGPN at limited label rates. Except for MLP, the remaining baselines are GNN-based models, and the detailed descriptions of these models are presented as follows:

(1) GCN [3]: As one of the most classic GNN models, it defines the graph convolution in the spectral domain and uses the first-order approximation to reduce the number of parameters.

(2) GAT [4]: It defines the graph convolution in the spatial domain by introducing the attention mechanism to assign different weights to the neighboring nodes when aggregating information.

(3) BGCN [5]: It views the observed graph as a realization from a parametric family of random graphs and targets inference of the joint posterior of random graph parameters and node labels.

(4) MVGRL [6]: It learns node and graph level representations by contrasting encodings from first-order neighbors and a graph diffusion.

(5) GPRGNN [7]: It aims to jointly optimize node features and graph topological information extraction by adaptively learning the Generalized PageRank weights.

Table 5: Classification accuracy of different methods with five labels per class

| Dataset | Cora | CiteSeer | PubMed | Photo |
|---|---|---|---|---|
| MLP | 41.58±4.11 | 40.33±5.38 | 60.36±3.50 | 55.63±4.29 |
| GCN [3] | 71.35±1.85 | 56.89±3.67 | 70.57±6.59 | 84.07±2.57 |
| GAT [4] | 72.40±2.45 | 61.31±2.29 | 70.86±4.63 | 82.88±2.71 |
| BGCN [5] | 74.54±2.58 | 61.25±5.06 | 70.18±5.72 | 79.91±4.74 |
| MVGRL [6] | 75.19±2.85 | 63.94±2.15 | 68.15±3.86 | 77.52±3.01 |
| GPRGNN [7] | 74.56±1.22 | 60.03±1.58 | 70.27±1.92 | 84.82±1.49 |
| APPNP [8] | 72.24±6.10 | 64.25±4.77 | 70.87±5.82 | 81.78±3.07 |
| CGPN-GCN | **75.83±1.69** | 65.00±2.59 | **71.20±4.05** | 84.77±1.90 |
| CGPN-GAT | 75.65±1.36 | **65.65±2.77** | 71.04±4.27 | **85.53±1.79** |

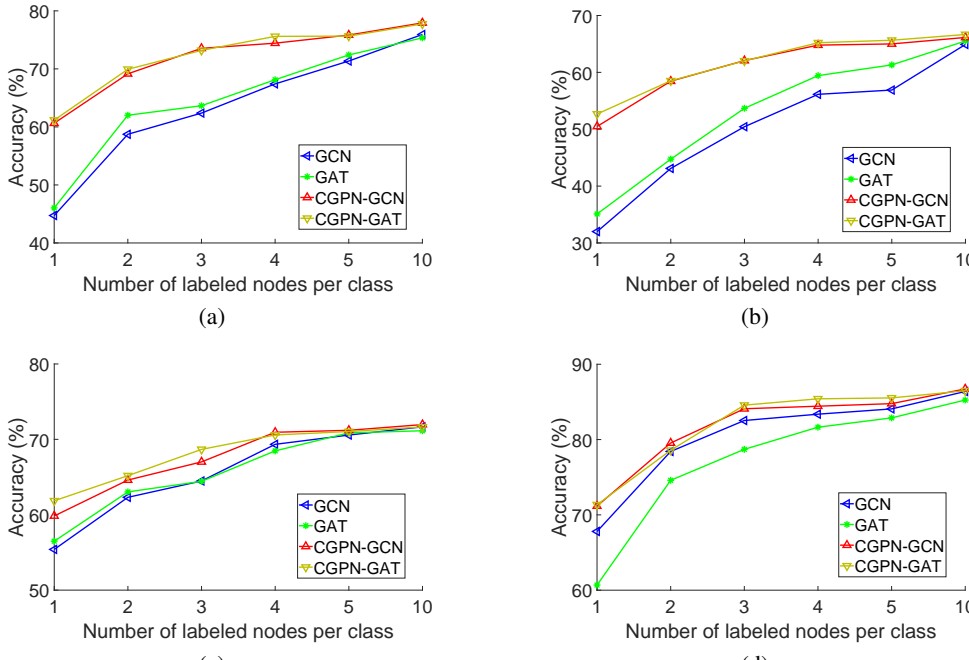

Figure 1: Classification performance with different numbers of labeled nodes. (a) Cora dataset; (b) CiteSeer dataset; (c) PubMed dataset; (d) Amazon Photo dataset.