# OpenReview forum: "Contrastive Graph Poisson Networks: Semi-Supervised Learning with Extremely Limited Labels"
_NeurIPS.cc/2021/Conference — NeurIPS 2021 Poster_

### Official Review · Reviewer_yD2p · 2021-07-13

**Rating:** 6
**Confidence:** 3

**Summary:**

For node classification tasks with extremely limited labeled data, this paper proposes a new GNN architecture GPN that extends Poisson learning to incorporate graph structure information and its contrastive learning method. The goal is to propagate the limited labels to the entire graph effectively to enhance classification performance. The framework consists of GNN for $p_\theta(Y|A, X)$ and GPN for $q_\phi(Y|A,X,Y_L)$ which are trained with ELBO loss and contrastive loss. GPN, inspired by GAN, is instantiated to learn attention-based edge weights used to construct a new diagonal matrix and graph Laplacian for Poisson learning.

**Limitations And Societal Impact:**

The negative social impact of this paper does not seem to be a concern.


**Main Review:**

The main experimental results and ablation study show the proposed method effectively propagates limited labeled data.
Using both variational loss and the contrastive loss for the output of GNN $p_\theta$ and GPN $q_\phi$ is novel and interesting.
After reading this paper, I found that the notion of propagating some information is similar to APPNP [1].  It would be nice to compare the performances of APPNP in the extreme case covered by this paper.
Additionally, the datasets on which this paper experiments are assortative so that propagating label information to the neighbors can be effective. Providing the experimental results on disassortative datasets such as Texas, Wisconsin, Actor, Squirrel, Chameleon, and Cornell would also be useful.


The followings are some questions for several unclear things.
- In Eq. (1) & (2), isn't  $u(x_i) \in \mathcal{R}^c$ and $L$ is an nxn matrix? How is their matrix multiplication defined?
- How many iterations we need for Eq. (7)? Doesn't it take a long time if large T is required?
- Ablation study for Eq. (8)
- Qualitative analysis for learned attention would be interesting

[1] PREDICT THEN PROPAGATE: GRAPH NEURAL NETWORKS MEET PERSONALIZED PAGERANK, ICLR2019

**Time Spent Reviewing:**

5

---

> ### Author Response · Authors · 2021-08-10
> **Response to Reviewer yD2p**
>
> We truly appreciate the constructive suggestions! Our clarification can be found as follows:
>
> **1. Re. performance comparison with APPNP [1]:** The performance comparison of APPNP with our proposed method is provided below, which shows that our CGPN-GCN outperforms APPNP with a large margin on all four investigated datasets when labeled nodes are extremely limited.
>
> Cora dataset
>
> | \# Labels per class | 1 | 2 | 3 | 4 | 5 |
> |  ----  | ----  |  ----  | ----  |  ----  | ----  |
> | APPNP | 53.52$\pm$12.05 | 62.07$\pm$4.46 | 62.02$\pm$9.34 | 70.92$\pm$3.79 | 72.24$\pm$6.10 |
> | CGPN-GCN | **60.64$\pm$9.18** | **69.15$\pm$9.03** | **73.54$\pm$2.76** | **74.43$\pm$2.25** | **75.83$\pm$1.69** |
>
> CiteSeer dataset
>
> | \# Labels per class | 1 | 2 | 3 | 4 | 5 |
> |  ----  | ----  |  ----  | ----  |  ----  | ----  |
> | APPNP | 47.94$\pm$10.46	| 56.59$\pm$9.22	| 58.61$\pm$11.04	| 62.34$\pm$5.47	| 64.25$\pm$4.77|
> | CGPN-GCN | **50.49$\pm$9.72** | **58.45$\pm$7.05** | **62.07$\pm$3.76** | **64.79$\pm$2.11** | **65.00$\pm$2.59** |
>
> PubMed dataset
>
> | \# Labels per class | 1 | 2 | 3 | 4 | 5 |
> |  ----  | ----  |  ----  | ----  |  ----  | ----  |
> | APPNP | 48.89$\pm$15.50	| 63.74$\pm$7.55 | 66.82$\pm$9.76	| 69.27$\pm$4.70 |	70.87$\pm$5.82 |
> | CGPN-GCN | **59.84$\pm$6.76** | **64.60$\pm$3.20** | **67.01$\pm$4.05** | **70.94$\pm$4.11** | **71.20$\pm$4.05** |
>
> Amazon Photo dataset
>
> | \# Labels per class | 1 | 2 | 3 | 4 | 5 |
> |  ----  | ----  |  ----  | ----  |  ----  | ----  |
> | APPNP | 67.11$\pm$8.40 | 70.03$\pm$11.51 | 78.99$\pm$3.23 | 81.17$\pm$4.03 | 81.78$\pm$3.07|
> | CGPN-GCN | **71.17$\pm$8.94** | **79.54$\pm$5.18** | **84.09$\pm$3.98** | **84.43$\pm$1.84** | **84.77$\pm$1.90** |
>
> **2. Re. results on disassortative datasets:** To further evaluate the effectiveness of our proposed method, we also provide the experimental results of APPNP [1], GPRGNN [2], and our CGPN-GCN on three types of disassortative datasets, namely the Wikipedia graph Chameleon, the webpage graph Cornell, and the Actor co-occurrence graph Actor. We use GPRGNN [2] as baseline method as it achieves the optimal performance among all the baseline approaches in our paper. We see that our CGPN-GCN is consistently better than APPNP [1] and GPRGNN [2].
>
>
> Chameleon dataset
>
> | \# Labels per class | 1 | 2 | 3 | 4 | 5 |
> |  ----  | ----  |  ----  | ----  |  ----  | ----  |
> | APPNP | 28.48$\pm$4.03 | 28.57$\pm$3.24 | 30.40$\pm$2.76 | 31.41$\pm$4.10 | 31.88$\pm$3.60 |
> | GPRGNN | 23.71$\pm$3.42	 | 28.28$\pm$3.59 |	29.27$\pm$4.52 | 31.69$\pm$3.18 | 32.60$\pm$3.88 |
> | CGPN-GCN | **29.35$\pm$2.46** | **32.00$\pm$2.31** | **34.42$\pm$2.21** | **35.10$\pm$3.23** | **36.87$\pm$2.60** |
>
> Cornell dataset
>
> | \# Labels per class | 1 | 2 | 3 | 4 | 5 |
> |  ----  | ----  |  ----  | ----  |  ----  | ----  |
> | APPNP | 32.40$\pm$22.83 | 38.99$\pm$15.42 | 41.71$\pm$15.70 | 43.11$\pm$14.19 | 45.09$\pm$14.21 |
> | GPRGNN | 33.69$\pm$17.85 | 39.47$\pm$10.56 | 43.15$\pm$13.23 | 45.09$\pm$16.14 | 50.25$\pm$12.29 |
> | CGPN-GCN | **51.69$\pm$5.23** | **54.70$\pm$3.44** | **56.03$\pm$2.08** | **57.02$\pm$1.87** | **57.46$\pm$1.61** |
>
> Actor dataset
>
> | \# Labels per class | 1 | 2 | 3 | 4 | 5 |
> |  ----  | ----  |  ----  | ----  |  ----  | ----  |
> | APPNP | 20.84$\pm$3.73 | 21.15$\pm$4.68 | 21.75$\pm$3.76 | 22.34$\pm$3.55 | 22.49$\pm$3.01|
> | GPRGNN | 18.61$\pm$5.21 | 19.77$\pm$3.77 | 21.53$\pm$3.23 | 22.56$\pm$3.48 | 23.31$\pm$2.27 |
> | CGPN-GCN | **21.04$\pm$2.93** | **22.85$\pm$1.87** | **23.99$\pm$1.58** | **24.35$\pm$0.97** | **24.92$\pm$0.51** |
>
> **3. Re. whether $u(\mathbf{x}_i)$ and $\mathbf{L}$ are matrices:** $u(\mathbf{x}_i)\in\mathbb{R}^{c}$ is the label vector of $\mathbf{x}_i$. $\mathbf{L}$ is the unnormalized graph Laplacian operator, rather than a matrix. To avoid such misunderstanding, we will correct $\mathbf{L}$ to $\mathcal{L}$ in our final version.
>
> **4. Re. iterations for Eq. (7):** In Eq. (7), we usually need around 10 iterations to produce satisfactory results. The computational cost is acceptable if a large $T$ is adopted, since the attention coefficients in Eq. (7) are shared across all the iterations, which means that the number of network parameters contained in Eq. (7) can be as small as that in a single-head graph attention network. As a result, the complexity of the Graph Poisson Network (GPN) (i.e., Eq. (7)) is $O(T(|V|FF’+|E|F’))$, where $F$ is the number of features, $F’$ is the number of hidden units, $|V|$ and $|E|$ are the numbers of graph nodes and edges, respectively. Empirically, we find that training the GPN (i.e., Eq. (7)) with $T=10$ only costs around 0.02s for each epoch on the Cora dataset containing 2708 nodes and 5429 edges. Therefore, the training of GPN (i.e., Eq. (7)) is quite efficient.
>
> **5. Re. Ablation study for Eq. (8):** We have conducted ablation study on the feature transformation module $f_{FT}$ used in Eq. (8), and the results below indicate that the performance of our CGPN-GCN will decrease without $f_{FT}$ (i.e., w/o $f_{FT}$) under different label rates.
>
> Cora dataset
>
> | \# Labels per class | 1 | 2 | 3 | 4 | 5 |
> |  ----  | ----  |  ----  | ----  |  ----  | ----  |
> | CGPN-GCN (w/o $f_{FT}$) | 57.72$\pm$6.32 | 63.69$\pm$6.07 | 69.37$\pm$4.83 | 70.55$\pm$2.50 | 72.98$\pm$0.72 |
> | CGPN-GCN | **60.64$\pm$9.18** | **69.15$\pm$9.03** | **73.54$\pm$2.76** | **74.43$\pm$2.25** | **75.83$\pm$1.69** |
>
> CiteSeer dataset
>
> | \# Labels per class | 1 | 2 | 3 | 4 | 5 |
> |  ----  | ----  |  ----  | ----  |  ----  | ----  |
> | CGPN-GCN (w/o $f_{FT}$) | 42.09$\pm$8.08 | 46.91$\pm$6.70 | 50.00$\pm$4.74 | 51.58$\pm$2.96 | 52.53$\pm$2.30|
> | CGPN-GCN | **50.49$\pm$9.72** | **58.45$\pm$7.05** | **62.07$\pm$3.76** | **64.79$\pm$2.11** | **65.00$\pm$2.59** |
>
> **6. Re. qualitative analysis for learned attention:** We show the average attention coefficients among all the seven classes on Cora dataset, where only five nodes are labeled in each class for training. Here, the attention coefficients have been normalized to $\[0,1\]$. Classes 1-7 are denoted as C1, C2, …, and C7, respectively. From this Table, it can be observed that our model tends to learn relatively large attention coefficients for intra-class nodes (see the diagonal elements). For example, in Classes 1, 3, 4, 5, 6, and 7, the intra-class attention coefficients are larger than most of the inter-class attention coefficients. This demonstrates that our Graph Poisson Network (GPN) does manage to discover the intrinsic relationships among graph nodes, which can help improve the prediction quality.
>
> |    | C1 | C2 | C3 | C4 | C5 | C6 | C7 |
> |  ----  | ----  |  ----  | ----  |  ----  | ----  | ----  | ----  |
> | C1 | 0.8532 | 0.9311 | 0.5928 | 0.8624 | 0.8252 | 0.6752 | 0.7474 |
> | C2 | 0.9311 | 0.6772 | 0.5358 | 0.8119 | 0.8514 | 0.5747 | 0.0000 |
> | C3 | 0.5928 | 0.5358 | 0.7933 | 0.7799 | 0.5803 | 0.8715 | 0.4534 |
> | C4 | 0.8624 | 0.8119 | 0.7799 | 0.9978 | 0.8639 | 0.7702 | 0.6906 |
> | C5 | 0.8252 | 0.8514 | 0.5803 | 0.8639 | 0.9404 | 0.9968 | 0.6886 |
> | C6 | 0.6752 | 0.5747 | 0.8715 | 0.7702 | 0.9968 | 1.0000 | 0.9363 |
> | C7 | 0.7474 | 0.0000 | 0.4534 | 0.6906 | 0.6886 | 0.9363 | 0.9545 |
>
> We also show the average attention coefficients of intra-class nodes on Cora dataset when only one, three, and five nodes are labeled in each class for training, respectively. Here, a growing trend of the intra-class attention coefficients can be observed with the increased number of labeled nodes. We infer that our GPN model can be gradually refined with the increase of label information, which helps recover the precise correlations among graph nodes.
>
> | \# Labels per class | 1 | 3 | 5 |
> |  ----  | ----  |  ----  | ----  |
> | C1 | 0.5539 | 0.6971 | 0.8532 |
> | C2 | 0.3745 | 0.4943 | 0.6772 |
> | C3 | 0.4853 | 0.6204 | 0.7933 |
> | C4 | 0.7676 | 1.0000 | 0.9978 |
> | C5 | 0.6775 | 0.8794 | 0.9404 |
> | C6 | 0.7145 | 0.9528 | 1.0000 |
> | C7 | 0.7077 | 0.9493 | 0.9545 |
>
> References:
>
> [1] Klicpera J, Bojchevski A, Günnemann S. Predict then Propagate: Graph Neural Networks meet Personalized PageRank[C]//International Conference on Learning Representations. 2018.
>
> [2] Chien E, Peng J, Li P, et al. Adaptive Universal Generalized PageRank Graph Neural Network[C]//International Conference on Learning Representations. 2020.

---

### Official Review · Reviewer_u74h · 2021-07-14

**Rating:** 4
**Confidence:** 4

**Summary:**

In this paper, authors propose the Graph Poisson Network (GPN) to  effectively propagate the information of limited labels to the entire graph, which is developed for semi-supervised node classification. The main contributions are as follows.

Firstly,  they integrate contrastive learning and variational inference framework into Poisson learning algorithm. This novel combination can propagate the label information more flexible.

Secondly, as they claimed, they achieve a significant improvement on semi-supervised node classification tasks.


**Limitations And Societal Impact:**

Yes

**Main Review:**

The motivation and method of this paper is interesting. They consider the graph neural network, variational inference, and the thought of contrastive learning simultaneously, and unify them with poison learning algorithm for better semi-supervised node classification tasks.

However, there are some problems in the paper that I think it still has a long way to publish in NeurIPS.

The first problem is writing. Although the authors use the mathematical views to describe their method, the notations are confused. For example, in Eq (1) in Sec. 2.3, what does $l$ and &n& means? It seems that they define them in 3.1, but I suggest that the authors can reorganize the mathematical descriptions for better reading. Also, what does $U$ in Eq. (3) means?

The second problem is the derivations. Eq. (4) is the most important loss in this paper. But the authors do not tell us how to derive the ELBO. You mean ELBO, I think you should point out what’s the ELBO? Does it the ELBO of label likelihood like $p(Y)$?

The third problem the contrastive label loss in Eq. (9). I didn’t find the definition of $z_i$ and $z_j$. Besides, the contrastive loss pull the predictions of different node pairs away. If different node pairs has the same label, does it make sense? I think you should pull the predictions of node pairs that has different labels.

The fourth problem is the experiments. I just see some simple numerical results about performance without any other explanations to show the advantages of their method. It is better to provide more insights about what does this model learn rather than simple numerical results.


**Time Spent Reviewing:**

2h

---

> ### Author Response · Authors · 2021-08-10
> **Response to Reviewer u74h**
>
> Thank you for your insightful comments! We have tried our best to address all your concerns.
>
> **1. Re. writing problem:** We will carefully reorganize the mathematical descriptions to avoid possible confusion on notations. In Eq. (1), $l$ indicates the number of labeled nodes, and $n$ indicates the number of all the labeled and unlabeled nodes. In Eq. (3), $\mathbf{U} \in\mathbb{R}^{n\times c} $ denotes the node labels generated by Poisson learning, where $c$ is the number of classes.
>
> **2. Re. derivation problem:** In statistical machine learning, maximizing Evidence Lower BOund (ELBO) is often used in Variational Bayesian (VB) methods to convert statistical inference problems (i.e., infer the value of a random variable given the value of another random variable) to optimization problems (i.e., find the parameter values that minimize an objective function). This technique can be found in many prior works, such as [1], [2], and [3].
>
> In our proposed method, we need to estimate the posterior $p_ \theta(\mathbf{Y}_ U|\mathbf{A}, \mathbf{X}, \mathbf{Y}_ L)$, where $\mathbf{A}$ is the adjacency matrix of the graph, $\mathbf{X}$ is the feature matrix of all the graph nodes, $\mathbf{Y}_ L $ and $\mathbf{Y}_ U$ indicate the label matrices of labeled nodes and unlabeled nodes, respectively. Here the posterior is often analytically intractable, and thus we resort to $q_ \phi (\mathbf{Y}_ U|\mathbf{A}, \mathbf{X}, \mathbf{Y}_ L) $ to approximate the true posterior. The ELBO has been derived in [4], [5], and [6], and it can be straightforwardly derived in our problem by following these existing works. Specifically, to derive ELBO loss, we first need to maximize the log likelihood of the observed labels $\log {p_ \theta }({\mathbf{Y}_ L}|\mathbf{A},\mathbf{X})$, which is equivalent to maximizing $\log \left( {{\mathbb{E}_{{q_\phi }({{\mathbf{Y}}_U}|\mathbf{A},\mathbf{X},{\mathbf{Y}_L})}}\frac{{{p_\theta }(\mathbf{Y}|\mathbf{A},\mathbf{X})}}{{q_\phi }({{\mathbf{Y}}_U}|\mathbf{A},\mathbf{X},{\mathbf{Y}_L})}} \right)$. Here $\mathbf{Y}$ denotes the label matrix  of all the nodes. According to Jensen’s inequality, we then have
>
> $&thinsp; &thinsp;&thinsp;  \log \left( {{\mathbb{E}_{{q_\phi }({{\mathbf{Y}}_U}|\mathbf{A},\mathbf{X},{\mathbf{Y}_L})}}\frac{{{p_\theta }(\mathbf{Y}|\mathbf{A},\mathbf{X})}}{{q_\phi }({{\mathbf{Y}}_U}|\mathbf{A},\mathbf{X},{\mathbf{Y}_L})}} \right)$
>
> $\geq {\mathbb{E}_{{q_\phi }({{\mathbf{Y}}_U}|\mathbf{A},\mathbf{X},{\mathbf{Y}_L})}}\left( {\log\frac{{{p_\theta }(\mathbf{Y}|\mathbf{A},\mathbf{X})}}{{q_\phi }({{\mathbf{Y}}_U}|\mathbf{A},\mathbf{X},{\mathbf{Y}_L})}} \right)$
>
> $= {\mathbb{E}_{{q_\phi }({{\mathbf{Y}}_U}|\mathbf{A},\mathbf{X},{\mathbf{Y}_L})}} {\log{{p_\theta }(\mathbf{Y}|\mathbf{A},\mathbf{X})}} - {\mathbb{E} _ {{q_\phi }({{\mathbf{Y}}_U}|\mathbf{A},\mathbf{X},{\mathbf{Y}_L})}}{\log{{q_\phi }({{\mathbf{Y}}_U}|\mathbf{A},\mathbf{X},{\mathbf{Y}_L})}}$
>
> $= \log {p_\theta }({\mathbf{Y}_L}|\mathbf{A},\mathbf{X}) - {\mathcal{D}} _ {KL} ({{q_\phi }({{\mathbf{Y}}_U}|\mathbf{A},\mathbf{X},{\mathbf{Y}_L})}||{{p_\theta }(\mathbf{Y} _ U|\mathbf{A},\mathbf{X})}) $
>
> $\triangleq {\mathcal{L}} _ {ELBO} (\theta, \phi)$
>
> Therefore, $\mathcal{L}_ {ELBO}$ is a lower bound of the log probability of the observed labels. In other words, maximizing the $\mathcal{L}_ {ELBO}$ is equivalent to maximizing the log probability of the observed labels. By maximizing $\mathcal{L}_ {ELBO}$, the true posterior can be approximated by $q_ \phi (\mathbf{Y}_ U|\mathbf{A}, \mathbf{X}, \mathbf{Y}_ L) $. To sum up, the $\mathcal{L}_ {ELBO}$ is a lower bound of the log likelihood of the observed labels.
>
> **3. Re. contrastive label loss:** (1) $\mathbf{z}_i$ and $\mathbf{z}_j$ denote the predictions on the nodes $\mathbf{x}_i$ and $\mathbf{x}_j$, respectively, which are generated from ${q_\phi }({\mathbf{Y}}|\mathbf{A},\mathbf{X},{\mathbf{Y}_L})$. Similarly, $\tilde{\mathbf{z}}_i$ and $\tilde{\mathbf{z}}_j$ denote the predictions on the nodes $\mathbf{x}_i$ and $\mathbf{x}_j$, which are generated from ${p_\theta }({\mathbf{Y}}|\mathbf{A},\mathbf{X})$. We will put the definition in a proper place. Meanwhile, to avoid possible confusion, Eq. (9) can be modified as follows:
>
> $ {{\cal L}_ {PC}} ({{\bf{z}}_ i},{\widetilde {\bf{z}}_ i})= \log \frac{ \exp (\left\langle {{{\bf{z}}_ i},{{\widetilde {\bf{z}}}_ i}} \right\rangle  /\tau)}  {\exp (\left\langle {{{\bf{z}}_ i},{{\widetilde {\bf{z}}}_ i}} \right\rangle /\tau ) + \sum\limits_{j = 1}^n {{\mathbb{1}_ {\[{\it j} \ne {\it i}\]}}\exp (\left\langle {{{\bf{z}}_ i},{{\widetilde {\bf{z}}}_ j}} \right\rangle /\tau )+ \sum\limits_{j = 1}^n  {\mathbb{1}_ {\[{\it j} \ne {\it i}\]}} \exp  (\left\langle {{{\bf{z}}_i},{{\bf{z}}_j}} \right\rangle /\tau )   } } $
>
> (2) Theoretically, the class-agnostic contrastive loss in Eq. (9) may indeed push some predictions of the examples in the same class apart, which can be referred to as biased contrastive learning [7]. Meanwhile, the contrastive objective that can simultaneously pull the intra-class predictions together and push the inter-class predictions apart is referred to as unbiased contrastive learning. However, according to [8], the bias between the two contrastive strategies can be gradually reduced with the increased amounts of classes and nodes, and their results can be very close with sufficient classes and nodes. Moreover, the biased contrastive learning is able to obtain promising results in multiple types of practical data such as images and graphs [9, 10, 11].
>
> Additionally, our proposed method mainly focuses on semi-supervised learning at extremely low label rates. As a result, the class information available for node discriminating is very limited. In particular, when there is only one labeled node per class, the unbiased contrastive learning can be equivalent to the biased contrastive learning. Nevertheless, we exhibit the results of the revised model that can pull the intra-class predictions together and push the inter-class predictions apart (denoted as CGPN-GCN (unbiased)) at different label rates on Cora dataset, and the results reveal that the models with different contrastive strategies achieve actually very similar results.
>
> | \# Labels per class | 2 | 3 | 4 | 5 |
> |  ----  | ----  |  ----  | ----  |  ----  |
> | CGPN-GCN (unbiased) | 70.05$\pm$7.70 | 72.62$\pm$3.88 | 74.04$\pm$3.24 | 75.47$\pm$1.52 |
> | CGPN-GCN (biased) | 69.15$\pm$9.03 | 73.54$\pm$2.76 | 74.43$\pm$2.25 | 75.83$\pm$1.69|
>
>
> **4. Re. the experiment problem:** Here we show that our method can obtain discriminative data representations due to the employed contrastive Graph Poisson Network (GPN). Specifically, we compute the average inter-class cosine similarity and average intra-class cosine similarity of the original features and the representations generated from contrastive GPN, respectively (see the tables below), where the values have been normalized to $\[0, 1\]$. It can be found that the representations produced by contrastive GPN generally reveal stronger intra-class correlations (see the diagonal elements) than the original features. Therefore, we believe that our contrastive GPN is beneficial to rendering promising classification results. In our final version, we will further use t-SNE method to visually show this effect.
>
> Average cosine similarity of the original features
>
> |    | C1 | C2 | C3 | C4 | C5 | C6 | C7 |
> |  ----  | ----  |  ----  | ----  |  ----  | ----  | ----  | ----  |
> | C1 | 0.5941 | 0.4486 | 0.1602 | 0.1509 | 0.1228 | 0.2443 | 0.3765 |
> | C2 | 0.4486 | 1.0000 | 0.2590 | 0.2390 | 0.1068 | 0.3458 | 0.3872 |
> | C3 | 0.1602 | 0.2590 | 0.6529 | 0.0721 | 0.0000 | 0.0948 | 0.1615 |
> | C4 | 0.1509 | 0.2390 | 0.0721 | 0.3391 | 0.0961 | 0.0761 | 0.0921 |
> | C5 | 0.1228 | 0.1068 | 0.0000 | 0.0961 | 0.3485 | 0.0481 | 0.0347 |
> | C6 | 0.2443 | 0.3458 | 0.0948 | 0.0761 | 0.0481 | 0.6061 | 0.2857 |
> | C7 | 0.3765 | 0.3872 | 0.1615 | 0.0921 | 0.0347 | 0.2857 | 0.6849 |
>
> Average cosine similarity of the representations generated by contrastive GPN
>
> |    | C1 | C2 | C3 | C4 | C5 | C6 | C7 |
> |  ----  | ----  |  ----  | ----  |  ----  | ----  | ----  | ----  |
> | C1 | 0.8746 | 0.5646 | 0.0000 | 0.3874 | 0.4075 | 0.6313 | 0.7609 |
> | C2 | 0.5646 | 0.8642 | 0.2530 | 0.4333 | 0.1417 | 0.3468 | 0.3709 |
> | C3 | 0.0000 | 0.2530 | 1.0000 | 0.2203 | 0.0377 | 0.2502 | 0.1517 |
> | C4 | 0.3874 | 0.4333 | 0.2203 | 0.7505 | 0.5081 | 0.1500 | 0.2666 |
> | C5 | 0.4075 | 0.1417 | 0.0377 | 0.5081 | 0.8946 | 0.4186 | 0.4269 |
> | C6 | 0.6313 | 0.3468 | 0.2502 | 0.1500 | 0.4186 | 0.8792 | 0.7070 |
> | C7 | 0.7609 | 0.3709 | 0.1517 | 0.2666 | 0.4269 | 0.7070 | 0.7636 |
>
> References:
>
> [1] Gershman S, Goodman N. Amortized inference in probabilistic reasoning[C]//Proceedings of the annual meeting of the cognitive science society. 2014, 36(36).
>
> [2] Kingma D P, Welling M. Auto-encoding variational bayes[J]. arXiv preprint arXiv:1312.6114, 2013.
>
> [3] Lopez R, Boyeau P, Yosef N, et al. Decision-Making with Auto-Encoding Variational Bayes[J]. NeurIPS, 2020, 33.
>
> [4] Bishop C M. Pattern recognition[J]. Machine learning, 2006, 128(9).
>
> [5] Yang X. Understanding the Variational Lower Bound[J]. Institute for Advanced Computer Studies. University of Maryland. Retrieved 20 March 2018.
>
> [6] Minka T. Divergence measures and message passing[R]. Technical report, Microsoft Research, 2005.
>
> [7] Chuang C Y, Robinson J, Lin Y C, et al. Debiased Contrastive Learning[C]//NeurIPS. 2020.
>
> [8] Saunshi N, Plevrakis O, Arora S, et al. A theoretical analysis of contrastive unsupervised representation learning[C]//ICML. PMLR, 2019: 5628-5637.
>
> [9] Chen T, Kornblith S, Norouzi M, et al. A simple framework for contrastive learning of visual representations[C]//ICML. PMLR, 2020: 1597-1607.
>
> [10] He K, Fan H, Wu Y, et al. Momentum contrast for unsupervised visual representation learning[C]//CVPR. 2020: 9729-9738.
>
> [11] You Y, Chen T, Sui Y, et al. Graph contrastive learning with augmentations[J]. NeurIPS, 2020, 33: 5812-5823.

---

> ### Author Response · Authors · 2021-08-27
> **Thanks a lot for your efforts in reviewing this paper**
>
> Dear reviewer u74h,
>
> Thank you again for your constructive suggestions.
>
> Your comments have enlightened us to think deeper and made our work more solid than before. We have carefully taken all the comments into consideration in the final version by providing detailed explanations and conducting more comprehensive experiments.
>
> We are wondering if our responses are helpful to solve your concerns. We would like to receive any further suggestions from you. We would be greatly appreciated that if you could reconsider the score considering all problems could be duly solved.
>
> Best,
> Authors

---

### Official Review · Reviewer_62ca · 2021-07-15

**Rating:** 7
**Confidence:** 4

**Summary:**

This paper focuses on semi-supervised node classification with extremely limited labels and the authors propose a new framework termed Contrastive Graph Poisson Networks (CGPN). Specifically, they design a Graph Poisson Networks to propagate the limited labels to the entire graph and employ a Graph Neural Network (GNN) to guide the propagation. Meanwhile, labeled data and unlabeled data are also considered in the final objective function. For the labeled data, they simply use a cross-entropy loss and for the unlabeled data, they apply a pairwise contrastive loss to explore extra supervision information.

**Limitations And Societal Impact:**

Yes

**Main Review:**

Pros:
1. Since few studies have focused on semi-supervised classification with extremely limited labels, this paper makes an interesting attempt. The authors propose Graph Poisson Networks (GPN) to avoid the performance degradation caused by the accumulation of inaccurate label prediction with iteration propagation.
2. The proposed method contains several specially designed parts to play different roles, which well handle the given data. This method is novel to me.
2. Experimental results demonstrate the effectiveness of the proposed method.
4. This paper is clearly written and well-organized.


Cons:
1. In section 3.3, the authors employ a feature transformation module $f_{FT}$ to modify the outputs of the last two iterations. However, it is not clear whether $f_{FT}$ works.
2. In line 181, "in the last two interations". Should it be “… last two iterations …”? The authors may further re-check the typos of this paper.
3. There are only tables on Page 8. The authors may avoid that by adding some words to this page.

Overall, I think this is a decent paper with a novel method that can be accepted.


**Time Spent Reviewing:**

5

---

> ### Author Response · Authors · 2021-08-10
> **Response to Reviewer 62ca**
>
> We are grateful for your valuable comments! The point-to-point responses are provided below.
>
> **1. Re. whether $f_{FT}$ works:** The feature transformation module is useful as it depicts the structural information formed by node features. We have also conducted ablation study on the feature transformation module on both Cora and CiteSeer datasets, and the results below indicate that the performance of our CGPN-GCN will decrease without $f_{FT}$ (i.e., w/o $f_{FT}$) under different label rates.
>
> Cora dataset
>
> | \# Labels per class | 1 | 2 | 3 | 4 | 5 |
> |  ----  | ----  |  ----  | ----  |  ----  | ----  |
> | CGPN-GCN (w/o $f_{FT}$) | 57.72$\pm$6.32 | 63.69$\pm$6.07 | 69.37$\pm$4.83 | 70.55$\pm$2.50 | 72.98$\pm$0.72 |
> | CGPN-GCN | **60.64$\pm$9.18** | **69.15$\pm$9.03** | **73.54$\pm$2.76** | **74.43$\pm$2.25** | **75.83$\pm$1.69** |
>
>
> CiteSeer dataset
>
> | \# Labels per class | 1 | 2 | 3 | 4 | 5 |
> |  ----  | ----  |  ----  | ----  |  ----  | ----  |
> | CGPN-GCN (w/o $f_{FT}$) | 42.09$\pm$8.08 | 46.91$\pm$6.70 | 50.00$\pm$4.74 | 51.58$\pm$2.96 | 52.53$\pm$2.30|
> | CGPN-GCN | **50.49$\pm$9.72** | **58.45$\pm$7.05** | **62.07$\pm$3.76** | **64.79$\pm$2.11** | **65.00$\pm$2.59** |
>
>
> **2. Re. “the last two interations”:** Thanks for pointing out this typo and we will carefully proofread our final version.
>
> **3. Re. There are only tables on Page 8:** Thanks for the suggestion and we will duly add some in-depth textual descriptions on the corresponding experimental results.

---

### Decision · Program_Chairs · 2021-09-27

**Decision:**

Accept (Poster)

**Comment:**

This paper addresses the issue of doing label propagation on a large graph where ground truth labels are extremely sparse.  The authors purpose a hybrid strategy using poisson networks and GNNs to learn a label propagation rule.  The reviewers agree that this is a well motivated and timely paper, however the original draft was light on experiments and the reviewers wanted to see a range of additional studies.  Several of these studies were provided during the rebuttal phase.  I have reviewed these studies myself, and I feel they add some insights to the paper and should be added in the camera ready.  There is still one remaining reject vote, but I think the criticisms were largely addressed in the rebuttal.  One criticism in particular, that the authors did not spent enough time introducing the ELBO, is reasonable but I do not consider it a fatal flaw.